# Measuring Terminology Consistency in Translated Corpora: Implementation of the Herfindahl-Hirshman Index

Angelina Gašpar [1] , Sanja Seljan [2,*] and Vlasta Kučiš [3]

1 Department of Philosophy, Catholic Faculty of Theology, University of Split, 21000 Split, Croatia; agaspar@kbf-st.hr
2 Department of Information and Communication Sciences, Faculty of Humanities and Social Sciences, University of Zagreb, 10000 Zagreb, Croatia
3 Department of Translation Studies, Faculty of Arts, University of Maribor, 2000 Maribor, Slovenia; vlasta.kucis@um.si
* Correspondence: sanja.seljan@ffzg.hr

**Abstract:** Consistent terminology can positively influence communication, information transfer, and proper understanding. In multilingual written communication processes, challenges are augmented due to translation variants. The main aim of this study was to implement the Herfindahl-Hirshman Index (HHI) for the assessment of translated terminology in parallel corpora for the evaluation of translated terminology. This research was conducted on three types of legal domain subcorpora, dating from different periods: the Croatian-English parallel corpus (1991–2009), Latin-English and Latin-Croatian versions of the Code of Canon Law (1983), and English and Croatian versions of the EU legislation (2013). After the terminology extraction process, validation of term candidates was performed, followed by an evaluation. Terminology consistency was measured using the HHI—a commonly accepted measurement of market concentration. Results show that the HHI can be used for measuring terminology consistency to improve information transfer and message understanding. In translation settings, the process shows the need for quality management solutions.

**Keywords:** terminology consistency; parallel corpora; translation; legislation; information transfer; Herfindahl-Hirshman Index (HHI)





## 1. Introduction

Research in the field of terminology evaluation has been active in recent years, either as a part of integrated projects or independent efforts, due to the relevance of terminology in multilingual surroundings or for information transfer, having the specific weight of carrying the meaning of a text. Consistent terminology is crucial to readability and comprehension. Ref. [1] stated that lexical cohesion is achieved by reiteration, repeated use of synonyms and hypernyms and collocations, and that "semantically heavy" words (especially noun phrases, followed by rare verbs) tend to be translated by literal translations in the specific genre of texts. Ref. [2] suggested several metrics to assess the suitability of the terminological resource with respect to a corpus by comparing the proportion of words in a lexical resource and corpus and the density of the average frequency of specific items. Bloomquist et al. (2021) [3] highlighted the need for consistent terminology in the medical domain, which is needed to improve communication with clinicians as choosing the right word or phrase to convey meaning eases readability and understanding, whereas Keloth et al. (2020) [4] studied terminological densities. Terminology was a key factor in transmitting information on disasters, as in McAleavy (2021) [5] and Christensen and Madsen (2020) [6]. Gottfried (2021) [7] and Pettinicchio and Maroto (2021) [8] researched the use of terminology on disabled persons to regulate medical rehabilitation and assess dimensions of disability.

Research on terminology variations and terminology consistency was performed in the domain of legislation in a multilingual setting, as in Pozzo (2020) [9]; in the domain of the crimila law, as in Komissarov and Komissarova (2021) [10]; for the designation of persons and professions, as in Kizil (2021) [11]; to define concepts of "extremism", as in Zhilina et al. (2021) [12].

Terminological consistency reduces uncertainty, ambiguity, or misunderstanding. Translation quality assurance, as defined by the EU DGT, refers to "correct usage of the target language, correct use of subject-specific and Community terminology, consistency with the original and between the different language versions, and compliance with the specific conventions for different types of texts (legal, political, letters, speeches, Web, etc.)". According to the European Commission's (EC) Studies (2012) [13], the EC invests millions of euros per year for the quality control of translations, training, IT, terminology tools, into correction of poor quality of external translations and correction of originals. Despite all efforts, discrepancies between language versions cause interpretation problems, mostly related to terminology. Terminological inconsistency denotes the use of two or more translation variants for a given source term. Consistent terminology can positively influence everyday and business communication, information transfer, and proper understanding. In multilingual written communication processes, challenges are augmented due to translation variants or missing terminology.

The main aim of this study is to implement the measurement metric for the terminology consistency, namely, to implement the Herfindahl-Hirshman Index (HHI) for the assessment of translated terminology in parallel corpora. We conducted research on a legislative corpus comprised of three types of legal subcorpora, dating from different periods and translated in different language pair directions. After the terminology extraction process, validation of the terminology candidates was performed, followed by an evaluation. Terminology consistency is measured by the HHI—a commonly accepted measurement of market concentration.

Although there are no specifications regarding recommended HHI scores, a higher score indicates a higher consistency and consequently leads to improved information transfer. For instance, in the process of benchmarking through diachronic analysis of documentation from a specific domain, the analysis of key terminology could reveal variations in terminology consistency. Another aspect could relate to the language analysis of specific products and services. Specific problems are faced in the multilingual environment or when searching for information by "language immigrants" or second language users, e.g., health information, as in [14], or legal information. In such situations, the HHI score could serve as a point of insight, yielding different interpretations due to terminology variations.

The rest of this article is organized as follows. In Section 2—Related Work, we present the results of studies on the relevance of terminology consistency for information transfer; in Section 3—Research, we present datasets used in the research, methodologies, as well as tools and principles for the HHI metric. In the Section 5—Results and Discussion, we present the results for some selected subcorpora, an analysis of the overall term structure, and a diachronic analysis of a selected subcorpus. Finally, conclusions and suggestions for further research are presented. The results indicate that the HHI can be used for measuring terminology consistency, influencing information transfer, and message understanding. However, the results are to be taken as preliminary results since a more detailed analysis on larger domain-specific terminology datasets is needed.

## 2. Related Work

Recently, information transfer based on terminology (in)consistencies has been a topic of interest in various domains, including legislation, medicine, agriculture, aquaculture, consumerism, and pandemics, among others. Consistent terminology contributes to correct information transfer and proper message understanding, which is a topic of interest in monolingual and multilingual settings.

Krauss et al. (2021) [15] presented an infrastructure based on findable, accessible, interoperable, and reusable principles for achieving semantic interoperability from biomedical terminology. They provided historical and current terminologies for not only a single language but also among various languages and transformed them into a health-network-compliant graph format. Bloomquist et al. (2021) [3] showed the need for uniform criteria to use consistent terminology in the medical domain, which was needed to improve communication with clinicians. There were variations in the accurate interpretation of immunostains, with a trend toward more accurate diagnosis, where the less-specific term "atrophic gastritis" was used in diagnostics. The study revealed deficiencies and variations in reporting practices, showing the need for precise and uniform terminology use. Keloth et al. (2020) [4] studied terminology densities to discover missing concepts and terminology candidates in a target language in the Unified Medical Language System through two source terminologies used in tandem and one target terminology. The research discovered 55 inconsistencies, afterwards resolved by an inter-annotator agreement of expert (39) and double quantity (98) of the additional concepts. Kachlik et al. (2020) [16] analyzed the need for uniform and clearly defined variant anatomy nomenclatures to improve understanding and diagnostic techniuqes, and Barnett (2021) [17] presented medical terminology that is commonly used in medical imaging.

Marshall et al. (2022) [18] investigated the effect of meat-related terminology on consumers (vegetarians) with regard to terminology-related concepts, i.e., preference for meat-related or neutral concepts.

Watson et al. (2021) [19] studied the use of terminology variants in the aquaculture domain, for the purpose of governance, environmental regulation, and licensing. The study results suggested that regulators and stakeholders need to agree on consistent terminology that characterizes the production environment. Rampasso and O'Grady (2022) [20] analyzed variations in the terminology of Hawaiian endemic species to facilitate species identification and proposed preferred terms.

Lanza et al. (2021) [21] performed a semantic comparative analysis between the current pandemic and the Spanish flu. The authors analyzed terms used to describe and communicate pandemic issues both to biomedical experts and to a nonspecialist public. The study was performed on two corpora on synchronic and diachronic levels to discover common lexical terminology in communicating pandemic issues.

Baumüller and Sopp (2021) [22] analyzed the change of terminology in the European accounting framework, from nonfinancial to sustainability reporting, showing the need for consistent company reporting and sustainability issues, which represents a considerable challenge of cost and practice in the EC.

Gottfried (2021) [7] presents the need for relevant legal requirements, definitions, and terminology related to the medical rehabilitation of disabled persons. Pettinicchio and Maroto (2021) [8] researched disability measurement across 65 countries and found that definitions, terminology, measurement, and instructions are relevant for understanding and assessing dimensions of disability.

Pozzo (2020) [9] argues that the EU law has a multilingual character which reflects multiculturality and which asks for the creation of a "neutral or descriptive" language, while legal terminology is often culture-bound, which is as great a challenge in translation. Komissarov and Komissarova (2021) [10] investigated the terminology of Ukrainian Criminal Law, arguing for more accurate and appropriate terms in order to improve accuracy, intelligibility, simplicity, and clarity for ordinary people, to increase the efficiency of criminal institutions, and to more effectively implement the concept of positive criminal liability in society. Kizil (2021) [11] analyzed British and American English juridical terms designating persons, such as representatives of different professions, persons with assigned juridical rights or duties, or criminals who break the law, especially terminology variants. Zhilina et al. (2021) [12] presented the approaches to the terminological characterization of the concept of "extremism" in legislative regulation, showing the lack of a united strategy. The authors performed a comparative analysis of the legislative regu-

lation of extremist crimes in foreign countries in order to identify common features and trends, legal responsibilities and differences and to develop unified concepts at the level of international legislation.

McAleavy (2021) [5] studied the importance of standardization and interoperability in the development of disaster management methods. In the context of disasters and catastrophes, there are large variations in norms, definitions, and language registers as well as variants due to diversity of culture, subcultures, and organizations in multilingual environments. Christensen and Madsen (2020) [6] highlighted the negative effects of inconsistent terminology, showing how they established a terminological ontology on incident management and how the results of terminology work may be stored in a term bank to support common understanding and efficient communication.

The terminology extraction of terminological variants has been the focus of many researchers who provide multiple perspectives on terminological variation, as in Drouin et al. (2017) [23], specifically the translation variation of multiwords, as in Araúz and Cabezas-García (2020) [24]; the description of terminological and phraseological features of legal genres, as in Ramos (2021) [25]; or terminological variation in specialized translation, as in Kerremans (2010) [26], Condamines (2010) [27], and Candel-Mora and Pastor (2012) [28]. Various studies have also focused on the usability of extraction tools, using statistical, linguistic, or hybrid approaches. Refs. [29,30] used combinations of extraction from corpora, including context, word alignment, and concordance, as in [31]. Software localization and industry classification was researched in [32]; extraction of ingredient names in nutrition was analyzed in [33]; extraction of paraphrases for technical medical terms was researched in [34].

Guillou (2013) [1] performed an analysis of statistical machine translation focusing on different parts of speech across different genres (technical text, novels, text on natural science) between English and French language pairs, showing that human translators mainly translate nouns rather consistently, although nouns do not always have the highest scores. Garcia et al. (2017) [35] proposed the integration of a new mechanism on a document level of machine translation to assess the lexical consistency of a translation and evaluated it using regular automated metrics.

A need for quality assurance was highlighted in [36] as part of the quality management system to deliver improved output (here translation) by giving an insight into the relevance of terminology management at a corporate level. Quality assurance is analyzed at the workflow, production, staff, users, software, and output levels.

Alwazna (2014) [37] emphasized the relevance of legal terminology in legal discourse, as well as the lack of translation methods. The author considered English-Arabic legal translation when coping with terminological incongruences. Ramuedzisi et al. (2019) [38] discussed the importance of language planning and policy, including principles for terminology development and management, deduplication, verification, authentication, and standardization).

Kwong (2021) [39] proposed a user-driven approach in assessing the performance of well-known term extractors, such as SDL MultiTerm Extract, using a benchmarking dataset of English–Chinese financial terms. The accounting terminology in the Croatian language can be perceived as complex or incomprehensible due to multiple changes and variations, as in Brozović et al. (2019) [40]. Mattila (2018) [41] delved deeply into a corpus analysis of linguistic variation in EU law.

Ramos (2021) [25] considered conceptual incongruity in intersystemic legal translation and the implications of ensuring harmonization and consistency in multilingual legal texts through institutional translation. A new perspective on textual terminology and terminological variation was proposed by Condamines (2018) [42]. Lapshinova-Koltunski (2015) [43] employed corpus-based methods to analyze translation variants in English to German translations produced with different translation methods. With an augmented amount of work, short deadlines, teamwork, and various digital resources, there is a growing need to ensure the quality of translation through the process and data analytics. For these reasons,

quality assurance (including terminology consistency checks) is suggested to become an integrative part of the Computer-Assisted Translation process [44].

## 3. Research

To implement the HHI for the assessment of translated terminology in parallel corpora, the research is performed on three types of legal corpora dating from different periods. The corpora are differentiated by methods of the translation process, such as the use of classical print resources or online open-source digital terminology bases, the number of translators (independent work or teamwork), and the use of information and communication technologies.

After the terminology extraction process, a validation of the term candidates was performed, followed by an evaluation. Terminology extraction was performed using several statistical tools and different methods, followed by a contrastive analysis among three types of subcorpora.

Terminology consistency was measured by the HHI for each subcorpus by selecting extracted terminology candidates having at least one terminological variant in the target language. The term selection criteria were based on the frequency of terms and their terminological variants.

### 3.1. Dataset

We used a 1.5 million-word Croatian–English parallel corpus of legislative texts comprising the subcorpora of Latin–English and Latin-Croatian subcorpus (Table 1).

**Table 1.** Corpora statistics.

| Parallel Corpora | English | | | Croatian | | |
|---|---|---|---|---|---|---|
| | **Pages** | **Tokens** | **Paragraphs** | **Pages** | **Tokens** | **Paragraphs** |
| Cro-Eng (1991–2009) | 1595 | 635,792 | 28,526 | 1356 | 506,338 | 25,416 |
| Canon Law (1983) | 119 | 123,875 | 5272 | 279 | 99,109 | 3639 |
| EU legislation (2013) | 160 | 81,413 | 2248 | 163 | 75,222 | 2447 |
| Total | 1874 | 841,080 | 36,046 | 1798 | 680,669 | 31,502 |

i.   Croatian-English parallel corpus (1991–2009)—Croatian legislative texts translated into English by 39 translators, as in Gašpar (2013) [45];

ii.  Canon Law corpus (1983)—English and Croatian versions of the Latin original text of The Code of Canon Law (CCL), available at http://www.arcc-catholic-rights.net/code_of_canon_law_1983.htm, accessed on 29 October 2021; https://zrno.wordpress.com/teoloske-teme/crkveni-dokumenti/zakonik-kanonskog-prava-1983/, accessed on 29 October 2021; the Croatian version was translated by 18 translators, as in Zec (2011) [46];

iii. EU legislation corpus (2013–)—English and Croatian versions of the EU legislation, comprising legislative texts (reports, protocols, agreements, regulations, working documents) from 2013, available at http://eur-lex.europa.eu/homepage.html, accessed on 29 October 2021.

The three corpora are the Croatian-English parallel corpus (1991–2009), Latin-English and Latin-Croatian versions of the Code of Canon Law (1983), and the English and Croatian versions of the EU legislation (2013).

### 3.2. Methods and Tools

After the process of harmonization and preformatting, sentence-level alignment was performed and translation memory was created using WinAlign 7.5.0.

Bilingual terminology extraction from the corpus was performed by SDL MultiTerm Extract at threshold frequency ($\geq$4). Croatian and English stop-word lists contained 1003 and 658 units, respectively. Details on terminology extraction were presented in [45].

Extracted terminology was evaluated using a manually created reference list comprising 4531 terms that were manually extracted from a legal subcorpus of 100,000 words. The evaluation was performed by a domain expert. It was based on the expert's subjective judgment, knowledge, and an assumption that recurrent multiword expressions are likely good terms. Among 10,000 validated terms, 100 of the most frequent terms having at least one terminological variant were selected and analyzed, as presented in Table 2.

**Table 2.** Extracted and validated terminology.

|  | Language Pair | Corpus Size | # Validated Extracted Terms | # Selected Terms |
|---|---|---|---|---|
| Cro–Eng (1991–2009) | Cro–Eng parallel | 1,142,130 | 10,000 | 100 Cro |
| Canon Law (1983) | Eng and Cro (from Latin) | 222,984 | 290 | 25 Cro 25 Eng |
| EU legislation (2013) | Cro-Eng parallel | 156,635 | 598 | 15 Cro 15 Eng |

Terminology extraction was performed from the two subcorpora: Canon Law—1983 and EU legislation—2013, using WordSmith Tools 5.0 to obtain keyword lists and concordances, frequency threshold ($\geq$5). SDL PhraseFinder tool was used for English multiword terminology extraction (Croatian language not supported) to obtain a list of the high-ranking term candidates. Croatian versions of both Canon law and the EU legislation were researched for the corresponding Croatian terms based on concordances obtained using the WordSmith tool. SDL MultiTerm Extract and WordSmith tools were used because they allowed for the context within which words or phrases were embedded. The context revealed relationships between words and made their meanings more evident. It would not be possible to infer the semantic properties of words from just frequency alone.

Among 290 validated terms extracted from the English version of Canon Law, 25 terms having at least one terminological variant were selected and analyzed. The list of extracted terminology from the English version of the EU legislation contained 598 validated terms, of which 15 terms were selected because of multiple-term variations, to assess terminological consistency, as presented in Table 2.

### 3.3. HHI

This research proposes an innovative use of the HHI to measure terminology consistency. It is used in the research because it can be used on a small amount of data and is easy to calculate. Because most terminology or keyword extraction tools are based on frequencies of terms or keywords, the integration of this simple measure could improve current quality assurance checkers. It was used by Itagaki et al. (2007) [47] to measure a terminology consistency according to the following equation:

$$\text{HHI} = \sum_{i=1}^{n} S_i^2 \tag{1}$$

where $S_i$ represents the market share of company $i$, and $n$ is the number of companies. If a single company dominates the entire market, HHI becomes 10,000 ($100^2$) or normalized to 10. If 10 companies have 10% each, the index is 1000 or normalized to 1. If two companies have 50% of the market share each, the HHI score is 5.0.

In this research, $i$ ranges over $n$ different translations for the specific term translated in the document. $S_i$ is the ratio of the number of times when the term was translated as

*I* to the total number of times it was translated. For instance, if a term has two different translations with equal frequency, the HHI score will be 5.0.

The HHI can be applied to the translation consistency check within specific company products, where the HHI score decreases if one source term has many translation equivalents. An overall translation consistency index ($C_t$) for a source term *t* is calculated as follows:

$$\frac{\sum\limits_{j=1}^{p} \sum\limits_{i=1}^{n} \left( \frac{f_i}{k_j} \cdot 100 \right)^2}{p} \tag{2}$$

where *p* is the number of translations having the source term *t*, and each frequency share is calculated as the ratio of its frequency $f_i$ to the total translation occurrence within a product $k_j$. HHI considers terminological variations and frequency for each term variation. It can be normalized to range between 0 and 10, where 10 indicates perfect consistency, i.e., one term in the source language is always translated by only one term in the target language. An example of the HHI calculation is shown in Table 3.

**Table 3.** Implementation of HHI.

| Source Language Term | Terminology Variants | Frequency | $\dfrac{\sum\limits_{j=1}^{p}\sum\limits_{i=1}^{n}\left(\frac{f_i}{k_j}\cdot100\right)^2}{p}$ |
|---|---|---|---|
| stečajno vijeće | insolvency tribunal | 61 | $C_t = 4.73$ |
| | bankruptcy tribunal | 31 | |
| | bankruptcy council | 7 | |
| | bankruptcy chamber | 11 | |

The consistency index is lower due to multiple terminological variants that lessen the term's "stability". The HHI for a hundred source language terms was calculated to measure interdocument consistency of a set of documents for 100 most frequent terms. This measure was based on term frequency counts of terms and their variants indicated to (in)consistent translations that significantly affected the quality assurance.

## 4. Results and Discussion

Results of the terminology consistency assessment for the three corpora are presented as follows.

### 4.1. HHI for Croatian–English Parallel Corpus

Among the total validated bilingual multiword terms extracted by SDL MultiTerm Extract, 100 terms having at least one terminological variant were checked for terminological consistency. Among the 100 terms, 43 terms had HHI scores lower than (<5); in other words, 57 terms had HHI scores higher than or equal to 5, $C_t$ (57) $\geq$ 5. The lowest HHI values were 1.5 and 1.6. The optimal HHI score ($C_t = 10$) was calculated for terms whose terminological equivalents were always the same, e.g., Cro. aktivna tvar/Eng. active substance; Cro. zaštita potrošača/Eng. consumer protection, and Cro. sukob interesa/Eng. conflict of interest. In the case of a dominant frequency of one terminological variant, the HHI score remained high and was slightly penalized, e.g., Cro. na temelju/Eng. on the basis of (freq. 171), based on (freq. 2), prescribed by (freq. 1), $C_t$ (na temelju) = 9.66.

The average terminology HHI for the 100 terms for terminology originally in Croatian and translated into English was 4.81, $C_t(100_{\text{Eng}}) = 4.81$, due to 321 terminological variants for 100 terms. The analysis of the histogram of 10 intervals for the Croatian–English (1991–2009) parallel corpus showed the highest frequency of consistencies was between 5 and 6, followed by consistency intervals 3–4 and 4–5, then 6–7, as presented in Figure 1.

The general score $C_t(100_{Eng}) = 4.81$ showed that, on average, for each term, there were at least two terminological variants.

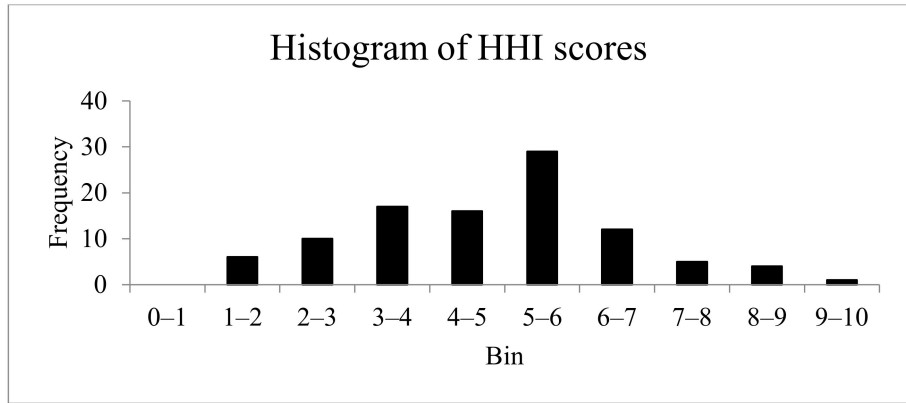

**Figure 1.** Histogram of the HHI scores showing the highest frequency for 5–6, followed by lower scores. The HHI scores >=7.0 which indicate the high level of consistency are at least represented.

The Figure 2 presents the average HHI for both language pairs for the EU subcorpus (1991–2009) showing slightly higher scores for the English–Croatian language pair $C_t(100_{Cro}) = 5.37$.

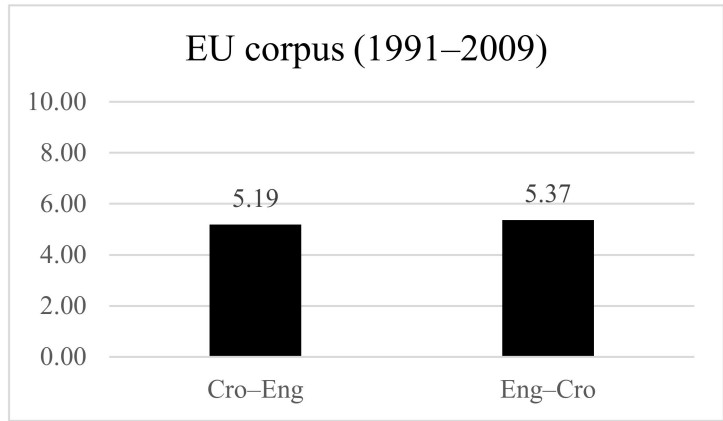

**Figure 2.** HHI scores for selected high-frequency terminology for the EU corpus (1991–2009) for two language pairs.

### 4.2. HHI for Canon Law Sample Corpus

The second subcorpus was the Latin–English and Latin–Croatian translations of the Latin original text of the Canon Law corpus (1983). The English keyword list was compared with the list of term candidates extracted by the tool PhraseFinder (the frequency threshold set to ≥3). Both lists showed frequency matching and the keywords appeared as headwords or modifiers of the extracted term candidates. Among 290 validated terms, 25 terms having at least one terminological variant were selected for the consistency check. The HHI scores for 15 target English terms were equal to or higher than 5.0, $C_t(15_{Eng}) \geq 5$, and the HHI scores for 15 target Croatian terms were higher than 5.0, $C_t(15_{Eng}) \geq 5$, of which 8 terms had a maximal score of 10.

The Figure 3. presents the overall HHI score for 25 selected terms for English translations was 5.09 $C_t(25_{Eng}) = 5.09$, and, for the same 25 terms in Croatian, it was $C_t(25_{Cro}) = 6.68$, probably due to native Croatian translators who translated the text from Latin.

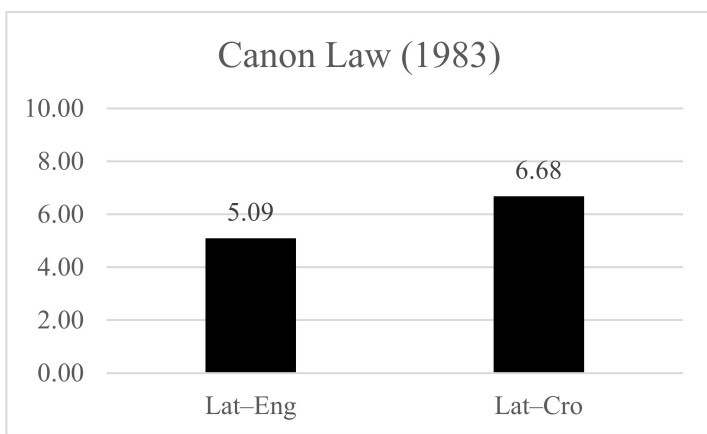

**Figure 3.** HHI scores for the translation terminology of the Canon Law (1983), showing higher scores for the translation into the Croatian language.

*4.3. HHI for EU Legislation Sample Subcorpus*

The third subcorpus was the EU legislation corpus (2013). A list of 598 validated terms was created, extracted by the PhraseFinder tool, and then compared to the English keyword list for frequency matching. Among 15 English terms, 9 had HHI scores equal to or higher than 5.0, $C_t(9_{Eng}) \geq 5$, whereas the corresponding 12 Croatian terms had HHI scores equal to or higher than 5.0, $C_t(12_{Cro}) \geq 5$, and, among them, two terms had the optimal score, 10.

The examples of inconsistent terminology are as follows:

- Eng. general provisions/Cro. opće odredbe, opći akti, uvodne odredbe;
- Eng. in accordance with/Cro. sukladno, u skladu s, prema;
- Eng. authorized representative/Cro. predstavnik, zastupnik, osoba ovlaštena za zastupanje, opunomoćenik;
- Cro. žalba protiv rješenja/Eng. appeal against the ruling, appeal lodged against a decision, appeal against the decision;
- Cro. rješenje iz stavka/Eng. decision referred to in/under paragraph, notice referred to in paragraph, ruling from paragraph, decision from paragraph, rules referred to in paragraph, formal decision from paragraph,
- Cro. laici/Eng. lay members, laymen, lay people, lay persons, etc.

Terms that have always been translated consistently:

- Eng. public institution/Cro. javna ustanova,
- Eng. religious community/Cro. vjerska zajednica,
- Eng. rights and freedoms/Cro. prava i slobode,
- Eng. legal person/Cro. pravna osoba,
- Eng. official seal/Cro. službeni pečat.

The Figure 4. presents the overall HHI score for 15 selected term translations, for the Croatian–English language pair $C_t(15_{Eng}) = 6.08$, and, for the same 15 terms for English–Croatian, it was $C_t(15_{Cro}) = 6.67$, for the EU corpus (2013–).

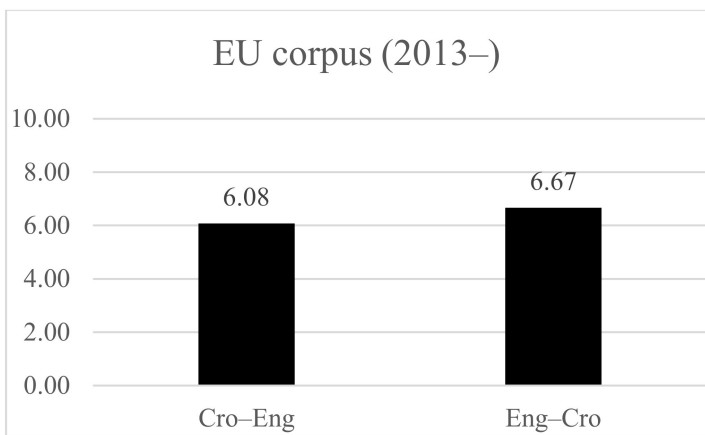

**Figure 4.** HHI scores for selected high-frequency terminology, showing better scores for English–Croatian language pairs (6.67).

*4.4. Analysis of the Overall Term Structure*

Owing to differences in the two languages, Croatian inflectional and English analytical ones, bilingual terms were analyzed considering their lexical and structural correspondences. Instead of cases, the English language uses prepositions, so it is interesting to examine how language differences affect terms' structures in both languages. As for translation consistency, translation variants, as the above examples suggest, are lexically and semantically different. Because legal language is characterized by context-specific terminology, these translation variants suggest that the inconsistencies in the source language negatively affect the target language's consistency. It also indicates the lack of common reference sources and multiple translators.

Multiword terms comprise several words but behave as single words. They are nonsubstitutional, noncompositional, and nonstandard sequences of syntactic categories. The importance of the grammatical relationships between their component parts motivated us to perform the part-of-speech (POS) structure analysis in assessing consistency.

The analysis of the most frequent POS patterns of 100 Croatian terms extracted from the first corpus showed the prevalence of noun phrases over prepositional and verbal phrases as follows: the most frequent was AN (31%), followed by NN (16%), NAN (11%), and NPrN (9%). The least frequent patterns were NA, VN, N, AAN, PrNPr, etc. The most represented n-grams were 2-g (freq. 56%), followed by 3-g (33%), 4-g (5%), and 1-gr (2%).

The analysis of the 25 Canon Law English terms resulted in the top-ranking POS pattern being AN (68%), followed by PrNPr (8%), NprN, VArtN, etc. N-gram values were highest for 2-g (68%).

The analysis of the 15 EU corpus English terms revealed that the most represented POS pattern was AN (freq. 33%). The most represented n-grams were 2-g (60%), followed by 3-g (26%). The extracted POS patterns showed that, in both languages, the AN pattern appeared as the most frequent, followed by the NN, NAN, and NPrN patterns. The POS patterns NA and VA exist in Croatian but not in English. In both languages, 2-g and 3-g were the most represented, as in [48].

The analysis of the N-gram correspondence (English–Croatian direction) revealed the top-scoring 2-g correspondence (37%), followed by the 3:2 (13%) and 4:3 (10%) correspondences. As the size of n-gram increases, the frequency of the relevant n-gram correspondence decreases. The use of prepositions and articles, which were expressed by in-flections in the Croatian language, increases the number of n-grams in the English language. Despite the language differences, the n-grams and matching patterns indicated compliance with the domain-specific stylistic rules concerning the use of legal terms in legislative texts (e.g., nominalization, domain- or subject-specific terms, phraseology, and syntactic features).

*4.5. Diachronic Analysis of the Croatian–English Parallel Corpus*

When analyzing the entire corpus from 1991–2009, most documents (97.8%) were created before the year 2006, when the Croatian Manual for Translating Legal Regulations into English was published to improve the quality of translations into the English language, suggesting the preferred terminology as in Novak (2006) [49]. After publishing the Manual in 2006, the number of terminology variants was reduced, e.g., the term "*predsjednik zastupničkog doma*" had two variants until 2006 ("speaker of the Sabor House of Representatives" and "president of the House of Representatives"), and after publishing the Manual it was reduced to "President of the House of Representatives". The term "prijelazne i završne odredbe" was reduced from four different terminological variants ("transitional and final provisions", "transitory and final provisions", "transitional and concluding provisions", and "Transitory and closing provisions") to one term, "Transitional and concluding provisions", used in the Croatian–English Parallel Corpus after the year 2006.

According to [45,50] at least 39 translators participated in the translation process. For our research purposes, two subcorpora were created to perform a diachronic analysis and to assess any improvement in the consistency of terminology:

- 10 documents created from 1991 to 2005 (before the publication of the Croatian Style Guide) and 10 documents from 2006 to 2009 (including 2006).

Detected inconsistencies in both subcorpora referred to the documents' titles; the names of institutions, ministries, and functions; and the use of punctuation, capital letters, and recurrent expressions, e.g., this Act shall come/enter into force; unless this act determines otherwise. In this process, 52 term candidates were selected for the analysis based on multiple terminological variants, of which 26 were found in the Croatian Manual, and, among them, 15 were selected for HHI analysis.

- The average HHI for the 100 terms of the entire corpus (1991–2009) was 4.81 for Croatian–English.
- The average HHI for 15 terms for 10 documents (1991–2005) translated from Croatian into English, before the Croatian Style Guide publication was 5.19.
- The average HHI for the same set of 15 term candidates, for 10 documents (2006–2009) after the Croatian Manual publication (2005) was 5.37.

The Figure 5. presents the improved HHI scores for the earlier EU corpus (1991–2009) and the later period subcorpus (EU corpus 2013–), after the accession of Croatia into EU, for both translation directions (Croatian–English and English–Croatian), and after the publication of the Manual in 2006, showing improved HHI scores. In both subcorpora, the English-Croatian translations showed better results of terminology consistency.

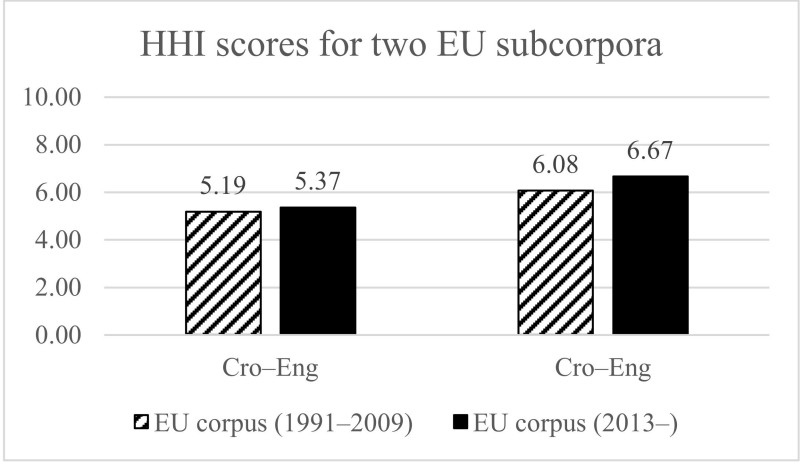

**Figure 5.** HHI scores for the later period subcorpus (EU corpus 2013).

## 5. Conclusions

This research aimed to implement the measurement metric for terminology consistency by the Herfindahl-Hirshman Index (HHI). This research was conducted on the legislative corpora, consisting of three types of legal domain subcorpora, dating from different periods: the Croatian–English parallel corpus (1991–2009), Latin–English and Latin–Croatian versions of the Code of Canon Law (1983), and English and Croatian versions of the EU legislation (2013). Consequently, we draw the following conclusions, as presented in Table 4.

**Table 4.** Summary of research results.

| Corpus | # Terms for HHI | HHI $\geq 5$ | Av. HHI for Target Language |
|---|---|---|---|
| Cro–Eng (1991–2009) | 100 Cro | $C_t (57_{Eng}) \geq 5$ | $C_t (100_{Eng}) = 4.81$ |
| Canon Law (1983) | 25 Cro<br>25 Eng | $C_t (14_{Eng}) \geq 5$<br>$C_t (15_{Cro}) \geq 5$ | $C_t (25_{Eng}) = 5.09$<br>$C_t (25_{Cro}) = 6.68$ |
| EU corpus (2013) | 25 Cro<br>25 Eng | $C_t (9_{Eng}) \geq 5$<br>$C_t (12_{Cro}) \geq 5$ | $C_t (25_{Eng}) = 6.08$<br>$C_t (25_{Cro}) = 6.67$ |
| 10 docs. 1991–2005<br>10 docs. 2006–2009 | 15 Cro<br>15 Cro | $C_t (8_{Eng}) \geq 5$<br>$C_t (12_{Eng}) \geq 5$ | $C_t (15_{Eng}) = 5.19$<br>$C_t (15_{Eng}) = 5.37$ |

When analyzing language pairs, HHI scores are higher for the English-Croatian language pairs, probably due to translation into the translators' native language and knowledge of the specific domain terminology.

When analyzing specific subcorpora, the highest HHI scores are obtained for the Canon Law Corpus (originally in Latin) translated into Croatian, with $C_t(25Cro) = 6.68$, and for the EU subcorpus (2013) translated from English to Croatian, with $C_t(25Cro) = 6.67$. The highest scores were possibly obtained because the translations were done by a smaller number of individuals (Latin Canon Law) or using online shared resources and the Croatian Manual for Translating Legal Regulations into English (2006).

The lowest HHI score was obtained for the Croatian one million word corpus (1991–2009) translated into English, $C_t(100Eng) = 4.81$, due to a larger number of translators having no common guidelines and terminological resources.

HHI scores were augmented for the Croatian–English language pairs when comparing periods before and after publishing the Croatian Manual in 2006, from $C_t(15Eng) = 5.19$ to $C_t(15Eng) = 5.37$, for the same specific terminology.

Our results support the possible implementation of the Herfindahl-Hirshman Index to assess terminology consistency, which could have a significant impact on information transfer and message understanding. This research was conducted on three different subcorpora, showing that inconsistencies in the legal genre occur despite the type of legal document, the number of translators, the use of digital or print language resources, or the time period.

The main limitation of this research is related to the analysis of different terminologies in different subcorpora, and the results are considered insights due to the relatively small terminology dataset, differentiating in the subcorpora. Future work would include a detailed analysis of larger sets of domain-specific terminology used in laws and regulations published after 2006 and comparisons of existing usage with officially published online resources, such as IATE, Eurovoc, and manuals prescribing the use of terminology. Another aspect of the research would include analysis of information transfer in the domain-specific documentation for the general public.

**Author Contributions:** The contributions of A.G. consist of data selection and conceptualization of the research. The contributions of S.S. consist of data analysis, formal analysis, supervision, and project administration. The contributions of V.K. consist of validation, investigation, and contextual analysis. All authors have read and agreed to the published version of the manuscript.

**Funding:** This research was funded by an institutional grant 11-929-1063 (Faculty of Humanities and Social Sciences Zagreb) supported by University of Zagreb.

**Data Availability Statement:** Corpora analyzed in the research were publicly available in the period of data collection.

**Conflicts of Interest:** The authors declare no conflict of interest.

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
