# Peer review of "Measuring Terminology Consistency in Translated Corpora: Implementation of the Herfindahl-Hirshman Index"

_information, doi:10.3390/info13020043_

Round 1
Reviewer 1 Report
This paper presents the Herfindahl-Hirshman Index (HHI) as a way to measure terminology consistency in translation variants across parallel corpora. This comes as an original idea in a somewhat under-researched area, but there is much room for improvement. First of all, the paper is not always clearly written or structured. Sometimes it is hard to follow the authors' train of thought, since sentences are not always linked to each other and most sections lack argumentation, especially those of Introduction and Related work.
References are unbalanced and not always updated. The authors should cite more relevant and updated work regarding terminology consistency and translation variants (e.g. Multiple Perspectives on Terminological Variation, by Drouin et al. 2017; Term and translation variation of multiword terms, by León-Araúz and Cabezas-García 2020). Rather than mentioning other works in long lists of references (e.g. lines 75-85), not always pertinent to the topic, the authors should consider giving a deeper insight into the state of the art and describing in greater detail the challenges posed by translation variants as well as the benefits of translation consistency.
Regarding the Methodology, the authors fail to explain a few things. For instance, why was the HHI chosen for this task? How could this improve current quality assurance checkers? How was the reference list of 4531 terms created? How was evaluation performed? Why did they use Multiterm Extract and not any other term extraction tool? They also state that they used WordSmith Tools (why not any more updated tool?) for terminology extraction, which makes hard to see the difference between the use of Multiterm Extract and that of WordSmith Tools.
The Research section could benefit from a graphic presentation of the results. Also, more examples of consistent and inconsistent translations would help to better understand the scope of the research and authors' understanding of the notion of consistency.
Given their size, the Research and Discussion sections could be merged into a single section called Results and discussion. Actually, the content of half of the Discussion section could go in the Introduction or be suppressed, since much of it is already covered in the Introduction.
The analysis of the overall term structure is not meaningful enough, especially if Croatian term POS patterns are not better contextualized. On the other hand, why is this analysis limited to POS structures (why not semantic or register-based analysis?) Why is it interesting enough in terms of consistency analysis? I wonder whether two slight morphological variants have the same implication as two cognitive or pragmatic variants in consistency measurement, or whether context should be manually analysed in order to find out whether inconsistency was deliberately sought. Actually, authors should clearly state from the very beginning how they understand terminology consistency and maybe delve into the motivations underlying term variation, which is not always an inconvenience.
In the Conclusions, the authors claim that this paper: (1) presents the challenges of translation consistency, such as a high number of translators, lack of time, pressure, etc. However, these are only hypothesis which are not proven throughout the paper; (2) performs an analysis of translated terminology, which is not thorough enough. Table 4 should not be included as part of the Conclusions but commented in the previous section (where actually found). No future research is included, although the authors state so in the Introduction.
References do not comply with any of the journal's references styles.
English should definitely be revised. The paper is not fluent enough and basic prepositions, articles, etc. are missing, which compromises comprehension.
For instance:
-Inaccuracies/confusing statements: in the abstract (and later on in the paper), authors state that the HHI index can be used for measuring terminology, when they actually mean measuring terminology consistency; in l. 37: authors should explain (or rephrase) what they mean by "single translations".
l.33: multilingual surroundingS
l.39: should THE terminological resource rather be terminological resources? Do they mean actual terminological resources or the terms employed in a translation?
l.44: typo: wrong inverted commas.
l.47: typo: Com-mission
l.52 needs rephrasing
l.55-56 articles (the/an) should be added.
l.64 Introduction par?
l.65 lack of consistency: part, section, chapter?
l.86: focusing of à focusing on.
l.89 nowà not?
l.180 vary-ants à variants
l.183: indicates to in/consistentà indicates in/consistent.
Etc.
Author Response
Dear Reviewer,
thank you very much for valuable comments. We are aware and agree with suggestions for the improvement and have put efforts to correct the paper according to your comments. All added text is marked in yellow. In the attachment file we provide answers, point by point.
Thank you and best regards,
Authors

Reviewer 2 Report
This paper deals with interesting issues and offers insights that put into question central concepts in terminology research. It is especially innovative the implementation of Herfindahl-Hirshman Index (HHI) for the assessment of translated terminology in parallel corpora
Author Response
Dear Reviewer,
thank you very much for the comment. We have made extensive language editing, performed by the professional editor, with specialization in computational linguistics and evaluation. The paper is formatted according to the journal style guide.
All added text is marked in yellow. In the attachment file we provide answers, point by point.
Thank you and best regards,
Authors,

Reviewer 3 Report
In general, this article is interesting and deserves to be published, however, there exist some non-critical downsides that should be removed.
- The reference list should correspond to the actual references in the article body. For example, the Ref. #15 (Melamed et al.) is not mentioned in the article.
- The reference list contains only three articles published in the last 5 years, where the two of them belong to one of the authors. The more recent papers should be added.
- The assumption that "HHI scores are higher for English-Croatian language pair, probably due to translation into native language and knowledge of the specific domain terminology" (lines 317-318) does not take into account the fact that the English language is generally rich in synonymy.
- The English presentation should be improved.
Author Response
Dear Reviewer,
We are very grateful for your comments. We have removed or added information according to your insightful suggestions. All added text is marked in yellow.
Thank you and best regards,
Authors,

Round 2
Reviewer 2 Report
After the incorporation of new data in the state-of-the-art part and the rest of the explanations requested by the reviewers, I consider the article ready for publication.
Reviewer 3 Report
Accept.